# Residual Stress and Tribological Performance of ZrN Coatings Produced by Reactive Bipolar Pulsed Magnetron Sputtering

**DOI:** 10.3390/ma14216462

**Published:** 2021-10-28

**Authors:** Anna Maria Laera, Marcello Massaro, Domenico Dimaio, Aleksandar Vencl, Antonella Rizzo

**Affiliations:** 1ENEA—Italian National Agency for New Technologies, Energy and Sustainable Economic Development, Brindisi Research Centre, S.S. 7, Km. 706, 72100 Brindisi, Italy; marcello.massaro@enea.it (M.M.); domenico.dimaio@enea.it (D.D.); antonella.rizzo@enea.it (A.R.); 2University of Belgrade, Faculty of Mechanical Engineering, Kraljice Marije 16, 11120 Belgrade, Serbia; avencl@mas.bg.ac.rs; 3South Ural State University, Lenin prospekt 76, 454080 Chelyabinsk, Russia

**Keywords:** dual magnetron sputtering, ZrN coatings, compressive residual stress, tribological properties, duty cycle

## Abstract

In the past few decades, ZrN thin films have been identified as wear resistant coatings for tribological applications. The mechanical and tribological properties of ZrN thin layers depend on internal stress induced by the adopted deposition techniques and deposition parameters such as pressure, temperature, and growth rate. In sputtering deposition processes, the selected target voltage waveform and the plasma characteristics also play a crucial influence on physical properties of produced coatings. In present work, ZrN thin films, obtained setting different values of duty cycle in a reactive bipolar pulsed dual magnetron sputtering plant, were investigated to evaluate their residual stress through the substrate curvature method. A considerable progressive increase of residual stress values was measured at decreasing duty cycle, attesting the significant role of voltage waveform in stress development. An evident correlation was also highlighted between the values of the duty cycle and those of wear factor. The performed analysis attested an advantageous effect of internal stress, having the samples with high compressive stress, higher wear resistance. A downward trend for wear rate with the increase of internal residual stress was observed. The choice of suitable values of duty cycle allowed to produce ceramic coatings with improved tribological performance.

## 1. Introduction

Over the last decades, protective tribological coatings have been strongly exploited in mechanical components for engines and transmissions, manufacturing industry tools, disk drives for the computer industry, precision instruments and human prostheses [1]. The life of the components and their cost are highly dependent on the performance and life of the selected coatings. Advanced physical vapor deposition (PVD) techniques are commonly used to engineer surfaces by customizing coatings with different materials including ceramics, among which titanium nitride (TiN) and zirconium nitride (ZrN) are the most used for their chemical stability, high melting point and their optical properties. Due to its high hardness, wear resistance, and pleasing light gold color, ZrN has attracted attention to improve the mechanical and tribological properties of conventional substrates in mechanical, optical, decorative and biomedical devices [2,3]. The increase in the life time of the tools with the application of ZrN layers, for example, is part of the strategy of saving critical materials present in the massive material of the same tool such us cobalt and tungsten [4].

A coated surface is defined by its functional parameters for elasticity, such as elastic modulus [5], for plasticity, such as hardness [6,7] or shear strength, and for ductility such as fracture toughness [8,9,10,11].

Although all coatings properties have been extensively studied, to deeply correlate the growth conditions with the residual stresses and the tribological performance remain a complex and not fully explored issue.

The residual stresses generated in thin films can be induced by external factors (extrinsic stress) or can be developed during the growing process (intrinsic stress). The most common external factor is the thermal mismatch that occurs in consequence of the cooling process after deposition, when the substrate and the coating have different coefficients of thermal expansion (thermal stress).

The more complex intrinsic stress results from contributions of a number of sources, some of which within the film bulk, such us impurity incorporation at lattice, voids, defect annihilation, phase transition. At the film/substrate interface, the main stress sources are the structural misfit between nucleated film and substrate and the intermixing processes. Finally, at the growing film surface the adsorption/desorption or the surface diffusion processes can have a significant effect on the resulting film stress.

The intrinsic stresses are inherently present in any coating deposited by the PVD process [12]. The stresses may be tensile or compressive and within the range of several GPa [13].

It is well known that the residual stresses may greatly affect the nanoindentation data of thin coatings and a linear increase of nanohardness as the function of residual stresses was usually reported [14].

The coating failure mode by cracking and by spalling depends both on the sign/magnitude of residual stress, and on the relative strengths of coating and coating substrate interface. It is generally accepted that, under tensile conditions, fractures normal to the interface develop starting from film defects and the resulting shear stresses along the interface may induce the coating de-adhesion. Under compressive stress, spallation may result either from the growth of a tensile wedge crack along the interface or by buckling and cracking of the coating [15].

It was also shown that by decreasing the residual stresses in the substrates, the adhesion can be improved, but few studies have analyzed in which way the residual stresses, related to film adhesion and mechanical properties, determine the wear mechanisms [16].

In the present work, ZrN thin coatings deposited on steel sheet were produced tuning the duty cycle in a reactive bipolar pulsed dual magnetron sputtering (BPDMS) system.

In bipolar pulsed sputtering, the target voltage is reversed and become positive during the pulse off period with the result to prevent the arc formation, to preserve a clean anode surface and to avoid hysteresis phenomena in the time/composition correlation of the film [17]. The suppression of arc events allows a more stable deposition process and extremely dense coatings with no discernible structural defects can be obtained. In spite of the large number of applications, many aspects of the complex BPDMS process are not yet well clear and in particular, it would be worth correlating the deposition parameters and the properties of deposited films. The choice of optimal values of pressure, temperature, growth rate, frequency, reverse voltage, duty cycle is crucial to enhance structural, mechanical, electrical, and optical properties of produced thin films. Numerous works in literature attested that each of these parameters play an important and complex role in the overall deposition process [18,19]. We focused our attention on the effect of duty cycle, defined as the ratio between the pulse length and the cycle time of the rectangular current or voltage pulses. The reduction of duty cycle induces the increase of the peak current and consequently plasma results in a richer ionic species [20].

The effect of duty cycle on the stoichiometry, microstructure, and mechanical proper-ties of ZrN coatings on steel sheets was previously discussed in ref. [21], where XRD analysis attested the cubic structure of deposited films. The duty cycle incisively affected lattice parameters and grain size, which reduced from 18.04 nm to 9.84 nm at decreasing duty cycle from 50% to 20%. In the same work [21], high values of hardness and indentation module were record for all produced coating with a maximum hardness of 31 GPa for the sample deposited at the lower value of 20% for duty cycle. In that work [21] the performed analysis allowed correlating the improved mechanical properties, obtained for duty cycles lower than 50%, to a coexistence of stoichiometric and over-stoichiometric (nitrogen rich) ZrN compounds. It has been reported in literature [22] that the over-stoichiometric cubic phase has a smaller volume and higher compressive stress with respect to the stoichiometric orthorhombic phase. The verification of this thesis passes through the study of the stress and the related tribological properties, that had been neglected in the previous work.

The residual stress was investigated through the substrate curvature method, by measuring the change of substrate curvature induced by the deposited thin film [23]. The obtained stress values were correlate to mechanical and tribological properties with the end goal to identify the optimal deposition conditions to produce reliable thin films. In industrial applications, it is critical to understand how the residual stresses in the coatings are related to the coating performance to extend the life of the components and reduce life cycle costs.

The acquired data demonstrate that stress have beneficial influence on produced ZrN coatings being the samples, with higher compressive internal stress more wear resistant.

Finally, it is shown that the films that exhibit higher compressive stress values are the same ones that had shown over stoichiometry [21] and consequently are the films that show optimal tribological and mechanical properties.

## 2. Materials and Methods

The ZrN films were prepared by bipolar pulse dual magnetron sputtering (BPDMS) system (KENOSISTEC KSA 75V, Milano, Italy), equipped with two rectangular Zr targets (99.9% purity) with dimensions of 406 × 50 mm, in a mixed atmosphere of Ar–N_2_, both 99.9999% pure. The partial pressure ratio of N_2_/(Ar + N_2_) was fixed at 0.1 and the gas working pressure was of 1 Pa. Two magnetron sources placed side by side were plugged to a generator in the symmetrical mode with a frequency of 80 kHz. A set of samples was prepared at constant power of 1.5 kW, averaged over the entire period, and different values of the pulse length (T) 12.5, 8.25, 6.25, 5.0 μs, respectively, as reported in Table 1

The distance between the sputtering target and the substrate was about 8 cm. The films were deposited at 300 °C on steel sheets substrate with a thickness of 0.4 mm and roughness R_a_ = 56 nm. The process chamber was warmed by using suitable heater resistors, and the temperature of 300 °C was monitored and kept constant using a thermocouple and a PID controller respectively (KENOSISTEC KSA 75V, Milano, Italy). A schematic diagram of the experimental apparatus was reported in [21]. A pulsed DC bias potential of −70 V at 50 kHz was applied to the substrate.

Previously calculated the rate, the deposition time was set in order to obtain the specimens with the same thickness as 1.7 μm.

The N_2_ flow was set so as sputtering occurred in metallic target mode, close to the onset of the instability region (hysteresis curve). This led to the formation of golden-yellow hue, highly crystalline, and close to stoichiometric ZrN films. Before each deposition process, the steel substrates were cleaned in an ultrasonic bath by using isopropanol.

Residual stress was investigated by the curvature method, i.e., a mechanical bending plate method (“thin foil method”). The mechanical profilometer (DektakXT, Bruker, Berlin, Germany) was used to determine the substrate curvature before and after deposition process. The measurements were repeated five times along the two main orthogonal axes for a distance of 15 mm on each sample in order to detect any inconsistences of residual stress. In this experimental set-up, the curvature resolution was 2 × 10^−4^ m^−1^. The curvature analysis was performed by using the Vision64 5.51 Bruker software (2015, Berlin, Germany).

The same profilometer was used to investigate the surface morphology through roughness measurements. A roughness Ra of about 60 nm was recorded for all deposited samples.

The pin-on-disk tribometer (Nanovea T500, Meteo-tech, Bnei Brak, Israel) was used to measure the wear rate and coefficients of friction in dry sliding conditions between a 100 Cr6 steel ball, having 6 mm in diameter, and ZrN coatings. Tribometer was used in circular sliding mode. All experiments were performed in air at a relative humidity of 60% ± 3% and at room temperature (23 ± 2 °C). The tests were performed at a sliding speed of 2 cm/s and the normal loads were 5, 10, 15 N on 25 m of length track, respectively.

Wear factors were calculated from the volume of material lost during testing. Although the coefficient of friction value was acquired from the contact of the ball and the coating over time, the wear was calculated by taking the cross-sectional area of displaced material at the end of test with the use of the profilometer [24].

## 3. Results and Discussion

### 3.1. Stress Measurements

The curvature method performed by using the profilometry is generally simple, reliable and economical but its application requires verifying a number of assumptions. The film and the substrate are taken as being homogeneous, isotropic, uniform in thickness and linear in elasticity. However, Stoney formula is often applied to practical cases where these assumptions are disregarded [25]. The starting deformation of the substrate is considered negligible and a perfect adhesion between film and substrate is also required. The thickness of the deposited film must be relatively low compared to the substrate. The average stress, assumed to be equibiaxial and laterally uniform, can be calculated by using the well-known Stoney equation [26]:(1)$$\sigma =\frac{1}{6}\left(\frac{1}{R_{\mathrm{post}}}-\frac{1}{R_{\mathrm{pre}}}\right)\frac{E_{s}}{\left(1-\upsilon_{s}\right)}\frac{t_{s}^{2}}{t_{f}}$$
where σ is the film internal stress after deposition, R_pre_ is the substrate radius of curvature before deposition, R_post_ is the substrate radius of curvature after deposition, E_s_ is the substrate Young’s modulus, υ_s_ is the substrate Poisson’s ratio, t_s_ is the substrate thickness, and t_f_ is the film thickness. The results of performed measurements, summarized in Figure 1, attested the crucial role of the target voltage waveforms on the growth process of ZrN thin films on steel substrates. All deposited films were characterized by presence of compressive residual stress with negative value of σ, that comprises the following two components:(2)$$\sigma =\sigma_{i}+\sigma_{\mathrm{th}}$$
where σ_i_ is the intrinsic stress and σ_th_ is the thermal stress due to the cooling down from the deposition temperature (300 °C) until room temperature. Being the thermal stress σ_th_ the same for all samples, the differences among σ values in Figure 1 can be ascribed to intrinsic stress developed during the deposition process. After growth interruption of ZrN films, a low mobility material, Abadias and al. [27] showed the absence of intrinsic stress relaxation process.

A considerable progressive increase of residual stress values was measured at decreasing duty cycle until 25%.

In sputtering deposition techniques, the compressive stress is generated when the incident particles, having energies from several tens up to 100 eV, impinge on film surface. The adatoms on the film surface can be knocked by the incident ions and in consequence become embedded into grains below the film surface, as describe in well-known atomic peening mechanism. These misfitted atoms, forced into small spaces, cause a lattice distortion and induce in surrounding matrix the strain field of compressive stress [28].

Energetic bombardment conditions also enhance adatom mobility and hence promote the diffusional incorporation of excess atoms in the grain boundaries, leading to a further increase in compressive stress, as recently proven by Magnfalt et al. [29,30]. A longer time between two successive impulses, at lower duty cycle, probably favors the adatoms migration before being impinged by other arriving adatoms. The grain boundary length scales inversely with the grain size. In consequence, the increase of stress magnitude at increasing inverse grain size, showed in Figure 2, supports the hypothesis that the compressive stress evolution involves grain boundary densification [29]. The error propagation theory was used to calculate the error of inverse grain size and the variance calculation to have a confidence interval of 95% for the stress value.

The stress development is concurrent to the formation of the over-stoichiometric (nitrogen rich) ZrN compounds found in ref. [21] and indicated by Chowalla et al. [22].

### 3.2. Tribological Properties

The results of the tribological experiments are shown in Table 2. Coefficient of friction was continuously measured, but only the values for the steady-state period are shown, i.e., values for the last 5 m were averaged. Wear volumes of the flat coated samples were calculated after each finished test, by measuring the wear track depth profile, according to ASTM G99. Since there were more different loads, the wear factor, i.e., specific wear rate values were calculated for comparison of the results. It was calculated as the units of volume loss (V) for the applied unit force according to the following relation:(3)$$W=\frac{V}{\mathrm{PL}}$$
where P is the applied normal load and L the sliding distance.

The values of the coefficient of friction (obtained in metal-ceramic contact) were in the range from 0.3 to 0.6, which correspond to the experimental values for metal-ceramic contact under dry sliding conditions [31]. The similar values of coefficient of friction were recorded for all deposited samples. However, a slight increase of the friction coefficient with increasing of normal load from 5 to 15 N was observed. The normal load during testing was relatively high, since it was a point Hertzian contact, which means that the asperity deformations were more plastic than elastic. It is well known that the friction depends on the nature of the asperity contact. If the contact between asperities is elastic, the coefficient of friction will decrease with the increasing pressure (adhesive component is decreasing and ploughing component is too small). On the other hand, if the contact between asperities is plastic, the coefficient of friction will increase with the increasing pressure (adhesive component is contact and ploughing component start to increase) [32].

Figure 3 shows the friction coefficient trend in function of sliding distance with an applied normal load of 5, 10, and 15 N.

For all samples, the coefficient of friction curves was similar, i.e., both running-in and steady-state period could be noticed [33]. The shape of all curves was common, i.e., it corresponds to the first type of the eight common types of friction curves [34]. The initial roughness of all samples was very low (Ra was approx. 60 nm), and after tests, it was much higher (order of magnitude of 1 μm, Figure 4). This suggests that at the beginning of the test high adhesion between coating and steel ball may be dominant and that in the initial (running-in) period roughing of the surfaces occur, followed by the intensive wear and formation of the wear products. This could be the explanation why the running-in period was longer for higher loads since usually, it is vice versa. According to the worn surface appearance (Figure 4) and wear factor values (Table 1) adhesion and deformation (ploughing) process dominated during the steady-state period. Fluctuation of the coefficient of friction values (deviation from the average value) was present throughout the examination, which suggests the occurrence of the micro stick-slip phenomenon (low sliding speed and high normal load cause high adhesion between asperities in contact) [35]. This was less pronounced at a higher load, since the ploughing (deformation) component of friction start to dominate.

Obtained wear factor values were in the range 10^−6^ to 10^−4^ mm^3^/Nm (Table 2). Literature data for sliding contacts under unlubricated conditions are from 10^−7^ to 10^−2^ mm^3^/Nm (for adhesive wear) and from 10^−5^ to 10^−1^ mm^3^/Nm (for abrasive wear) [36]. Wear factor is a very useful engineering tool, since it is approximately equal for certain load intervals [36] and could be easily determined from the Equation (3). Three wear regimes could be distinguished (shaded differently in Table 2). At low loads and low duty cycles, adhesive wear was dominant, while at higher loads and higher duty cycle, share of the abrasive wear start to increase [37]. As commonly observed for ceramic materials, the grain-boundary microcracking occurred. When the generated microcracks intersect each other, fine particles of material are detached from the surface and carried away from the contact area as wear debris [38,39,40]. Analysis performed with the profilometer confirmed the presence of the typical signs of adhesive and abrasive wear mechanism, in which adhesion, cracks and single grain pull-outs cause the material removal. The 3D image for wear track of the most worn sample ZrN-A and the cross-section of the wear profile of sample ZrN-B in Figure 4, clearly show the presence of material flow and scratches together with a great amount of wear debris accumulated in the wear track.

As reported in Table 2, for each value of applied normal load the wear rate decreased at decreasing duty cycle. The graph in Figure 5 exhibits the downward trend for wear rate measured with an applied normal load of 5 and 10 N. The sample ZrN-C and ZrN-D, deposited respectively with a duty cycle of 25% and 20%, had high wear resistance with an applied load of 5 N. Depth of the abrasive scratches was much higher than the grain sizes, so the grain-boundary strengthening could not have the effect, i.e., complete grains were detached during scratching.

### 3.3. Properties and Performance Relationships

As evaluated in a previous work [21], all the obtained ZrN coatings exhibited a cubic structure, and the transition from a strong columnar texture along (111) direction, present in sample ZrN-A, to a more disordered mixed columnar structure along (111) and (220) direction, at deceasing duty cycle was observed. The grain size decreased from 18.04 nm to 9.84 nm decreasing the duty cycle, for ZrN-A and ZrN-D, respectively, and coatings having small grain size resulted with higher compressive stress and hardness.

These data are in clear departure from results of Tung et al. [19] who describe an opposite trend between grain size and hardness values for ZrN coatings obtained by using filtered cathodic arc ion-plating system. These discrepancies can be explained considered that the compressive stress in ZrN coatings, described in Tung’s work, was mainly due to lattice defects introduced through atomic peening mechanism, while in present work the grain boundary densification was also identify as a significant compressive stress source. In addition, the range of grain size values considered in Tung’s work was significantly lower, being between 4.9 and 7.9 nm. In this region below 10 nm, the inverse Hell Petch behavior is usually observed [41,42].

In present work, the samples hardening was detected at reducing grain size from 18.04 to 9.84 nm, following the classical Hall–Petch effect, that enhances the possibility of the dislocation blocking with the piling up of dislocation at grain boundaries [43].

Either the compressive stress increase or the grain size decrease may be factors that affect the hardening of coatings.

The tribological behavior is also influenced from grain size, hardness, and compressive stress. The ZrN-C and ZrN-D samples, obtained selecting a duty cycle of 25% and 20%, turned out to have not only the highest stress values and hardness, but also the greatest wear resistance. Influence of the compressive residual stress was more pronounced for lower loads, i.e., when adhesive wear was dominant regime. In that case Hall–Petch strengthening through the yield strength increase could have an effect. At higher loads and abrasive wear regime, brittle nature of ceramic coatings diminishes the influence of grain-boundary strengthening. A seemingly linear correlation between residual compressive stress and wear rate was found for lowest normal load, as showed in Figure 6.

These data provide evidence of the BPDMS huge potential to produce thin ZrN coatings having large wear resistance. For comparison, we summarize in Table 3 some selected examples published in the last 10 years, in which hardness, stress values and, in some cases, wear, were evaluated in ZrN thin films deposited through alternative PVD techniques. Although the BPDMS allowed to obtain ZrN coatings having inferior hardness values respect to high power pulsed magnetron sputtering (HIPIMS), the lower wear values recorded in present work attested the significant tribological performance of produced coatings. However, a direct comparison of wear values reported in Table 3 was not possible since tribological tests were performed by using the same applied load but with spherical indenters made of different materials.

## 4. Conclusions

The BPDMS was used to deposit ZrN coatings on steel sheets selecting different values of duty cycle in the range from 50% to 20%. All the internal stresses of produced thin films resulted compressive and were quantify through the substrate curvature method.

The obtained results indicated that compressive residual stress significantly depend on selected voltage waveform. The highest values of compressive residual stress were obtained applying a duty cycle of 25% and 20%. The increase of residual stress was concurrent to the reduction of grain size. In addition to atomic peening mechanism, the grain boundary densification also plays a role in compressive stress evolution.

The combined analysis of microstructure and residual stress was crucial to understand the influence of inherent properties on tribological behavior of ceramic coatings. The highest values of compressive residual stress were related to high hardness values and low wear value. A linear correlation between residual compressive stress and wear was found. The typical signs of adhesive and abrasive wear mechanisms were identified.

These findings can help to design and to produce ceramic coatings with properties tailored for specific practical use. The choice of a suitable value of duty cycle allowed to tune the compressive residual stress and, in consequence, the mechanical and tribological properties of deposited ZrN coatings. Although excessive stress can induce deadhesion, with film buckling or cracking, the presence of compressive stress can have a beneficial effect on hardness and wear resistance of ZrN thin films.

## Figures and Tables

**Figure 1 materials-14-06462-f001:**
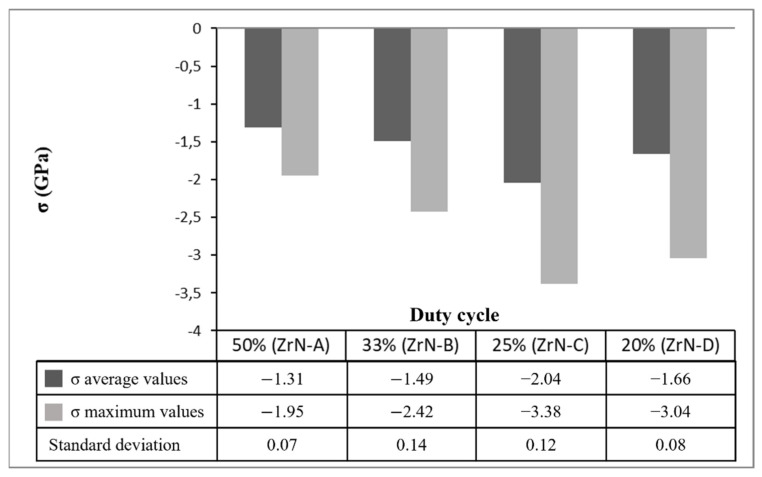
Average, maximum values and standard deviation of compressive residual stress of ZrN films deposited with different duty cycle. The samples name is included in brackets.

**Figure 2 materials-14-06462-f002:**
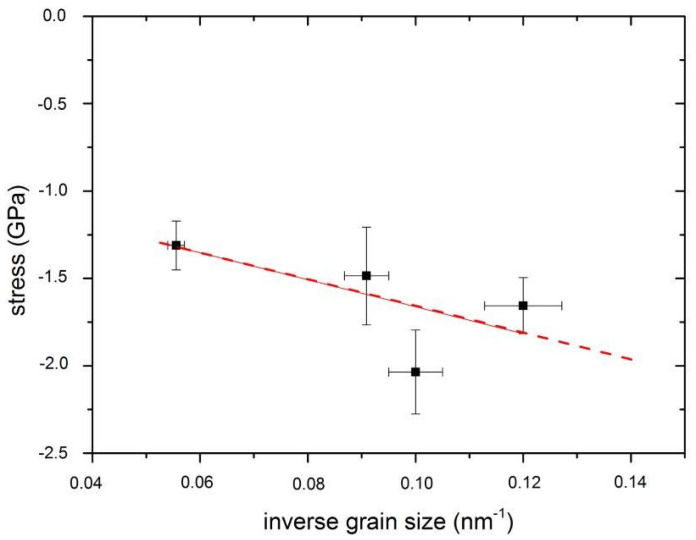
Compressive stress as a function of inverse grain size.

**Figure 3 materials-14-06462-f003:**
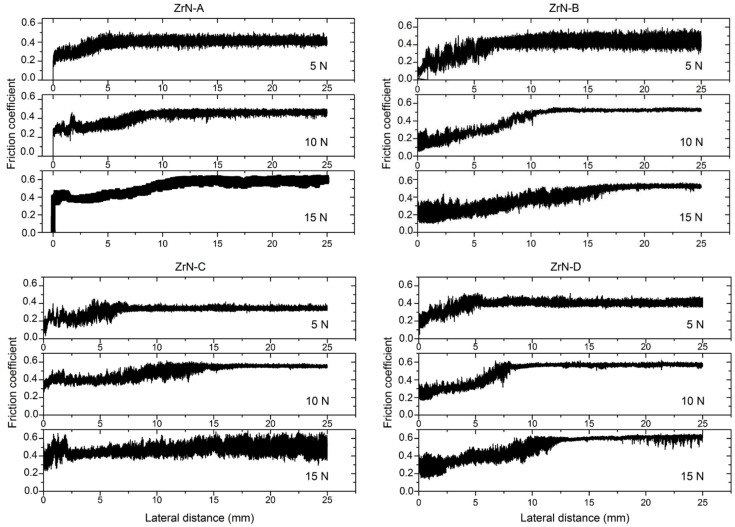
Friction coefficient measurements at loads of 5, 10, and 15 N.

**Figure 4 materials-14-06462-f004:**
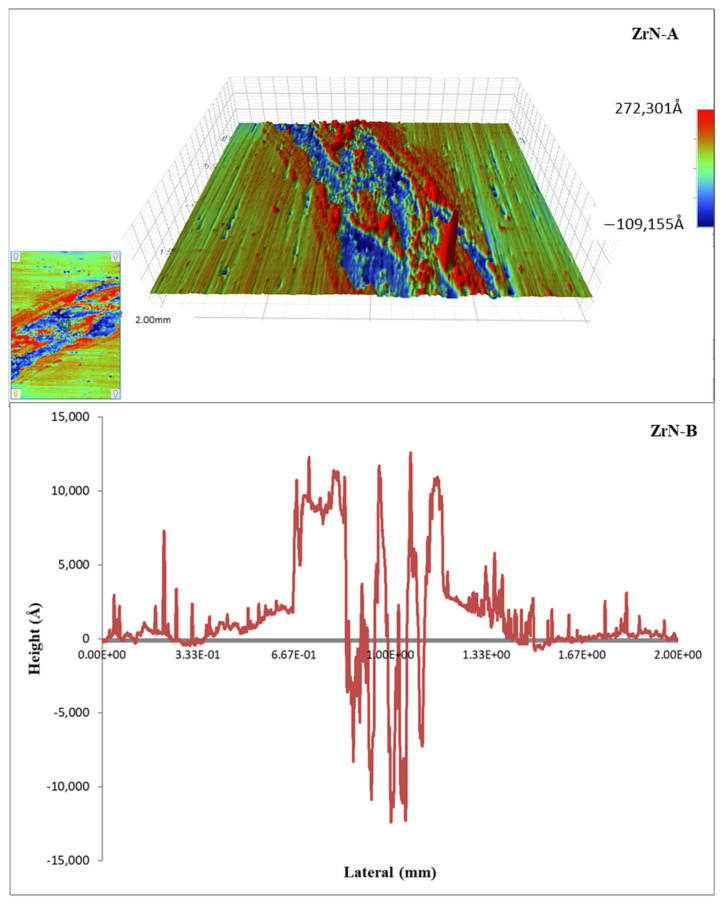
Surface morphology of scratch track of sample ZrN-A and wear profile of sample ZrN-B at normal load of 10 N.

**Figure 5 materials-14-06462-f005:**
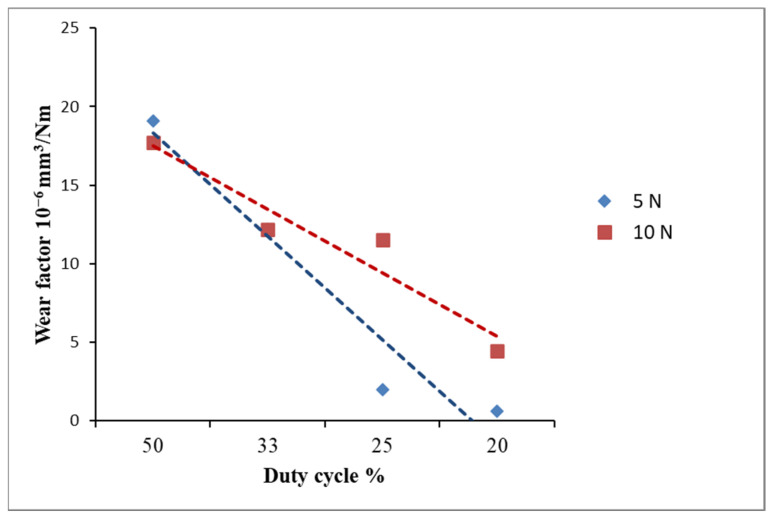
Wear factor as a function of applied duty cycle.

**Figure 6 materials-14-06462-f006:**
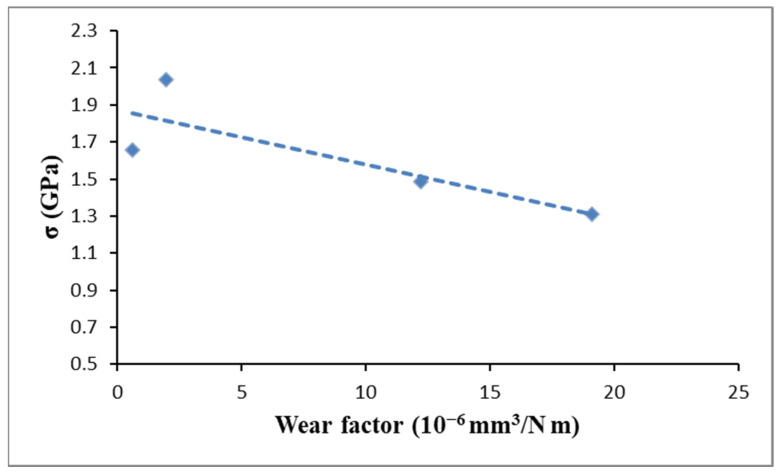
Correlation between factor rate and internal compressive stress measured applying a load of 5 N.

**Table 1 materials-14-06462-t001:** Deposition parameters and thickness of deposited of ZrN films.

Sample	Duty Cycle	T (μs)	Voltage (V)	Current (A)	Thickness (μm)	Pulse Power (W/µs)
ZrN-A	50%	12.5	460	3.23	1.46	118.864
ZrN-B	33%	8.25	549	2.74	1.53	176.971
ZrN-C	25%	6.25	614	2.40	1.75	235.776
ZrN-D	20%	5.0	693	2.15	1.59	297.99

**Table 2 materials-14-06462-t002:** Coefficient of friction and wear factor measured applying normal load of 5, 10, and 15 N at sliding speed of 2 cm/s.

Sample	Coefficient of Friction			Wear Factor 10^−6^ mm^3^/Nm		
	5 N	10 N	15 N	5 N	10 N	15 N
ZrN-A	0.41	0.45	0.52	19.10	17.70	174.90
ZrN-B	0.46	0.45	0.47	12.20	12.20	11.20
ZrN-C	0.33	0.54	0.46	1.98	11.50	11.20
ZrN-D	0.40	0.57	0.62	0.60	4.46	9.42

**Table 3 materials-14-06462-t003:** Grain size, thickness, stress, hardness, and wear factor of ZrN thin films deposited through alternative types of sputtering methods, recently published, in comparison to the results obtained in present work.

Deposition Technique	Grain Size (nm)	Thickness (µm)	Stress (GPa)	Hardness (GPa)	Wear Factor (10^−6^ mm^3^/Nm)(load 5 N)	Ref.
BPDMS	9.84	1.59	−1.7	31	0.6	[21] this work
Magnetron Sputtering	19	1.1	−4.2	34	-	[41]
	11.59	1.5	−0.7	23	-	[3]
	39.5	1.2	−3.3	30	-	[44]
Cathodic Arc	18	2.5	−5.8	37	8.01	[45]
	7.9	0.9	−13	30	-	[19]
HIPIMS + Magnetron Sputtering	9.1	2.5	−4.1	35	5.33	[45]
HIPIMS	-	1.84	−10	40	6.84	[46]
		1.96	−7.7	36	6.37	
		2.13	−5.1	32	7.19	
	-	0.58	−4.2	27	-	[47]

## Data Availability

Not applicable.

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
