# Peer review of "Residual Stress and Tribological Performance of ZrN Coatings Produced by Reactive Bipolar Pulsed Magnetron Sputtering"

_materials, 2021, doi:10.3390/ma14216462_

Round 1

Reviewer 1 Report

The article is essentially devoted to the analysis of the relationship between internal stresses in niobium nitride films and their microstructure. The reviewer believes that grain size data from their previous work should be briefly described. One of the reasons for this wish of the reviewer is the metrological aspect of the 18.04 nm record (line 140).

The way in which curvature measurements and internal stress calculations are reported also requires metrological details.(1) The lateral dimensions of the specimens and the base of the curvature measurement are not indicated.(2) It is necessary to provide detailed data on the datascattering, and not only the maximum and average values.And again, the record, for example, the average voltage of 1.311 GPa (Fig. 1) is not correct, in the opinion of the reviewer.

The reviewer believes that presenting the stress measurement statistics will give a clearer conclusion about the relationship in Figure 2.

Also the reviewer has a few suggestions

(1) In formula (1) indicate the index s of Young's modulus and Poisson's ratio. As indicated in Ref. 24.

(2) lines 169-171. Please check the correctness of the phrase.

In paper 24, the reviewer read, in his opinion, a statement on a similar topic.

Ref 24, Section 2, 1st paragraph: As no stress evolution was observed during growth interrupts at 270 C (see Fig. 2 of Ref. 13), any stress relaxation mechanisms can be ruled out for these low-mobility materials;

Reviewer 2 Report

I found the received article interesting and undoubtedly worthy of publication. Unfortunately, I think it needs to be corrected and reorganized.

I noticed that the article is a continuation of the research described in reference 22. Unfortunately, this mention appears as late as in the Results and discussion section. The fact that it adds to the knowledge provided in Ref. 22 with updated material is not a harmful matter, and I do not see any obstacle in publishing it. Nevertheless, I would like to suggest to the authors to modify the text. I have tried to group my suggestions into the following list:

  1. Abstract in my opinion is not detailed enough. It makes minimal reference to the results of the experiment. Considering that readers usually finish reading the article at the abstract, the authors risk a worse citation index. Their experiment, research, and conclusions should be more emphasized.
  2. The first part of the Introduction section is, in my opinion, well written. I feel that the purpose of the study is vaguely presented. It is entirely unclear why the authors choose such a magnetron technique with variable duty cycle to study CrN layers with tribological methods. I would suggest to the authors that the continuation of the previous research (Ref. 22) should be the aim of this paper. Moreover, this goal is not accidental. The previous research dealt with the material characterization of ZrN produced under specific and established technological conditions. The current experiment concerns the investigation of the functional properties of this material and is complementary to the previous one.
  3. The Experimental section suffers significantly from the lack of description of any technical details. I was able to understand the experiment only after reading Ref 22. However, I could only learn about it from the later section. Undoubtedly, it is worth considering supplementing the description here with the technology or explicitly referring to the experiment in Ref.22. If I were the authors, I would include the schematic of the apparatus and the characteristics of the power pulses in the device used with the particular indication of the duty cycle variation parameter. Of course, this would have to be a different schematic than in Ref. 22 and a schematic representation of the power pulse waveform for copyright reasons. In my opinion, such a step would make the article easier to read for readers.
  4. How were the substrates prepared? Were they polished? Were they etched?
  5. How was the temperature controlled? Was 300 ℃ set intentionally from an external heat source, or were the samples heated by plasma exposure?
  6. If the power pulse length varies from 5.0 - 12.5 µs, how was the power kept constant at 1.5 kW?
  7. The "Results and discussion" section, lines 121 - 143 - This should, in my opinion, be part of the Introduction and include a starting point for formulating the objectives of the paper. Furthermore, the article's aim should be to continue the research and complement it with the utility value - anti-wear properties.
  8. I encourage the authors to supplement the text thus modified with numerous references to the material characterization results of Ref.22 in the "results and discussion" section. I acknowledge that the authors have done this in several fragments, but there are some left without material interpretation, for example, discussing the results in Fig.1. In my opinion, the SEM images from Ref. 22 directly illustrate the effects of energetic plasma particles on the structure of the layers and thus fit perfectly with the authors' interpretation.

To sum up, I feel that referring explicitly to the connections of the current material with previous studies and discussing them with the previous article will make the article more coherent.

Reviewer 3 Report

It is a interesting work on ZrN coating. But I have some comments on this manuscript.

  1. About residual stress measurement, the curvature method was used in this manuscript. Just as author said, "its application requires verifying a number of assumptions. The  film and the substrate are taken as being homogeneous, isotropic, uniform in thickness and linear in elasticity."  What is the substrate for residual stress measurement? Is it steel sheets?  If the substrate is the steel,  the substrate may have plastic deformation with 1-3GPa stress.  This methods can not be used for  plastic deformation sample.
  2. Fig.1,  unit should be added. MPa?
  3. For the wear resistance, the Friction coefficient is importment. Coefficient of friction curve of all the sample should be described and discussed.
  4. The films adhesion also is importment for films wear resistance. So the authors also should discuss the relationship of the films stress and films adhesion.
  5. The films surface morphology also should be described and the surface morphology also have importment role on films friction.

Round 2

Reviewer 2 Report

he authors have responded to all my questions and made the proper amendments in the text.

Reviewer 3 Report

It can be accepted.